# Datavzrd: Rapid programming- and maintenance-free interactive visualization and communication of tabular data

**Felix Wiegand**[1]*, **David Lähnemann**[1,2], **Felix Mölder**[1,3], **Hamdiye Uzuner**[1], **Adrian Prinz**[1], **Alexander Schramm**[4], **Johannes Köster**[1]

1 Bioinformatics and Computational Oncology, Institute for AI in Medicine (IKIM), University Hospital Essen, University of Duisburg-Essen, Essen, Germany, 2 German Cancer Consortium (DKTK), Partner Site Essen-Düsseldorf, A Partnership Between DKFZ and University Hospital Essen, Essen, Germany, 3 Institute of Pathology, University Hospital Essen, University of Duisburg-Essen, Essen, Germany, 4 Laboratory of Molecular Oncology, West German Cancer Center, Department of Medical Oncology, University Hospital Essen, Essen, Germany

* felix.wiegand@uni-due.de

**Data availability statement:** Code is available under https://github.com/datavzrd/datavzrd.

## Abstract

Tabular data, often scattered across multiple tables, is the primary output of data analyses in virtually all scientific fields. Exchange and communication of tabular data is therefore a central challenge. We present *Datavzrd*, a tool for creating portable, visually rich, interactive reports from tabular data in any kind of scientific discipline. *Datavzrd* unifies the strengths of currently common generic approaches for interactive visualization like R Shiny with the portability, ease of use and sustainability of plain spreadsheets. The generated reports do not require the maintenance of a web server nor the installation of specialized software for viewing and can simply be attached to emails, shared via cloud services, or serve as manuscript supplements. They can be specified without requiring imperative programming, thereby enabling rapid development and offering accessibility for non-computational scientists, unlocking the look and feel of dedicated manually crafted web applications without the maintenance and development burden. *Datavzrd* reports scale from small tables to thousands or millions of rows and offer the ability to link multiple related tables, allowing to jump between corresponding rows or hierarchically explore growing levels of detail.

## Introduction

Tabular datasets, prevalent across diverse scientific fields such as biology, physics, economics, and environmental science, stand as the primary outcomes of scientific investigations. Communicating such results in a way that others can access the contained information at ideally the same level of detail as the original analysis author is a major cornerstone of reproducibility and transparency *after* publication. Moreover, it is important for the efficiency of scientific

**Funding:** The author(s) received no specific funding for this work.

**Competing interests:** The authors have declared that no competing interests exist.

communication *before* publication, enabling non-computational scientists to, for example, dive deeply into data underlying high-level figures.

A traditional option for such communication are spreadsheet applications like *Excel* (and their corresponding file format xlsx, in the following for simplicity called *Excel* files) which is still commonly regarded as a primary choice, especially among non-computer science professionals. Most spreadsheet applications, including *Excel*, lack the capability to quickly and easily visualize different columns of a dataset. They also do not support reproducibility of results for similar datasets without repeating the work. Finally, multiple investigations of supplementary Excel files show that the automated conversion of certain values can lead to the misinterpretation of data: for example, gene names such as *SEPT2* (Septin 2) are inadvertently converted into dates, leading to errors in genomic research [4,7].

Beyond plain spreadsheet representations, other solutions are available. Naturally, any general purpose programming language (e.g. Python or R) or transformation languages (e.g. XSLT) could be used (e.g. in comparison with helper libraries like great-tables [26]) to implement entirely custom reporting. Moreover, various frameworks have been developed that further simplify these tasks. *Shiny* [9] works by implementing the intended visuals and corresponding interfaces in an R or Python script, which is then executed in a server process that provides the resulting user interface in the form of served HTML pages. ShinyApps.io [27] enables easy hosting of *Shiny* applications without the need to maintain an own web server. However, at the time of writing, it is limited to five applications, 25 usage hours per month, and 1 GB per dataset with the free plan. Similarly to Shiny, *Dash* [28] allows creating interactive dashboards with Python code, but also requiring a running server process. *Lumen* [29] provides data-driven dashboards that can be configured without requiring imperative programming, using the declarative YAML language, but also requires a server process. Of course, charting libraries such as D3.js [30] or Plotly [31] enable the creation of interactive visualizations, but using them within tables requires substantial bespoke coding and web development effort. MultiQC [2] enables the summarization of various standard formats into single-page HTML-based reports. Unlike Dash, Lumen, and Shiny, MultiQC reports are portable and do not require a running web server. Apart from pre-implemented standard formats, MultiQC also offers the ability to specify custom data, however, without being able to interactively explore large interrelated complex tables.

Finally, there are numerous specialized web applications that aim to make certain domain specific kinds of data explorable [14–18]. Those applications often combine query interfaces with tabular and plot based visualizations, backed by a server process that extracts information from a database. Their development is complex and requires thousands of lines of code and work hours, although they share similar visuals, and likewise require substantial workload for maintenance of the installation. Finally, data security requirements often turn their exposure to the public into a complex and sometimes even impossible task, so that they cannot be easily used to communicate the results supporting a publication in a transparent way.

So far, there was no solution available that would maintain both a visual and interactive interface to any kind and size of tabular data as well as maintenance-free, reliable, and sustainable availability guarantees.

To overcome this situation, we present *Datavzrd* (https://datavzrd.github.io), which aims to provide visually rich, interactive, and portable representations of tabular data, while avoiding the caveats with above listed approaches. Datavzrd can be used in virtually any kind of data analysis for any kind of tabular data type in a rapid and ad-hoc fashion, obviating the need for specialized implementations and continuous maintenance as well as deployment of a web application.

*Datavzrd* is implemented as a command line application with the *Rust* [32] programming language. It is available as an MIT licensed open source software via *GitHub* [33], can be installed via *Cargo* [34] and *Conda* [35], or used as a *Snakemake* wrapper [36] for rapid integration into reproducible data analysis workflows.

## Results

### Feature overview

Without requiring imperative programming, *Datavzrd* can produce highly interactive and visually rich interfaces (cf. Fig 2) from tabular input files in various formats (CSV, TSV, Parquet [37] or JSON). Using simple YAML-based declarative specifications, users define datasets along with the desired visualizations for each column. The self-contained report is then generated by executing *Datavzrd* with the command `datavzrd path/to/config.yaml -o path/to/output`. Within the YAML specifications, a plethora of visual standard elements can be activated per table column, including tick plots (Fig 11A), bar plots (Fig 11B), heatmaps (Fig 11C), pill plots (Fig 11D), ellipsis, custom plots, linkouts (Fig 11E), and scientific notation of floating point numbers. The resulting per-cell visualizations are complemented by extensive annotation and control over the displayed tables, including sorting, filtering and searching (Fig 9), optional line number display, per-column histograms, additional header rows, table descriptions, and the ability to hide columns into a detail-mode.

Beyond single tables, it offers the ability to jump back and forth between corresponding rows in multiple related tables and to create custom visualizations that entirely replace the default table views with Vega-Lite [3] plots or plain HTML. Importantly, *Datavzrd* reports are encoded as standalone HTML files that do not require a web server for viewing them and can be simply opened in any HTML5 [38] compliant web browser. This way, they can be easily shared via Email or attached as supplementary files to publications. Additionally, it also eliminates server maintenance and reduces server load by shifting processing load to the browser (without hampering scalability, see section Scalability).

For enhanced accessibility and ease of hosting, *Datavzrd* provides a `publish` subcommand, which automates the upload of reports to a newly created or existing GitHub

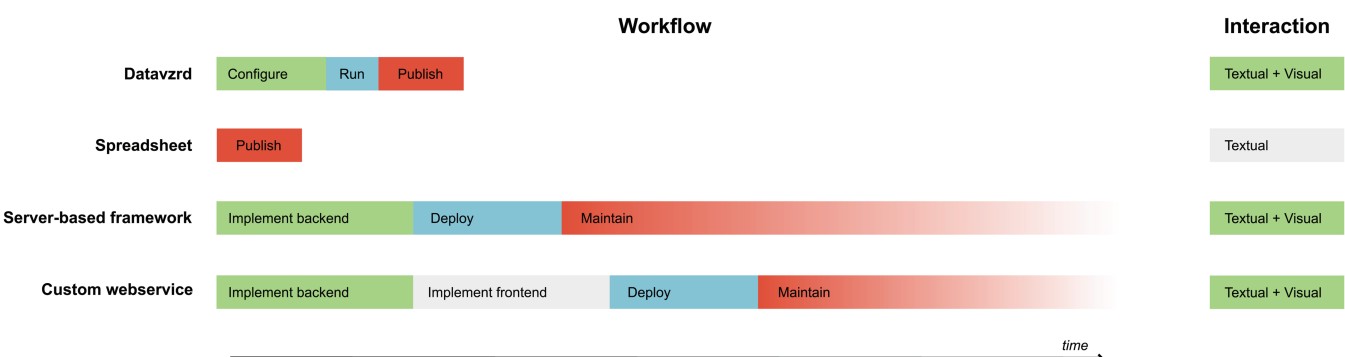

**Fig 1. Timeline comparison.** Comparison of the work items needed for publication and communication and the interaction capabilities of *Datavzrd* compared to above mentioned alternative approaches. Time per step is meant as a relative approximation of the true times, which may of course vary depending on the actual kind of data that shall be handled. We postulate that, for example, simple configuration requires less time than backend or frontend implementation and running a command line tool like *Datavzrd* is less effort than deploying a web service on a given infrastructure. Finally, the required effort for *Datavzrd* ends with the publication of the generated static HTML files, whereas server based approaches require continuous maintenance of their deployment in order to remain secure, operable, and accessible.

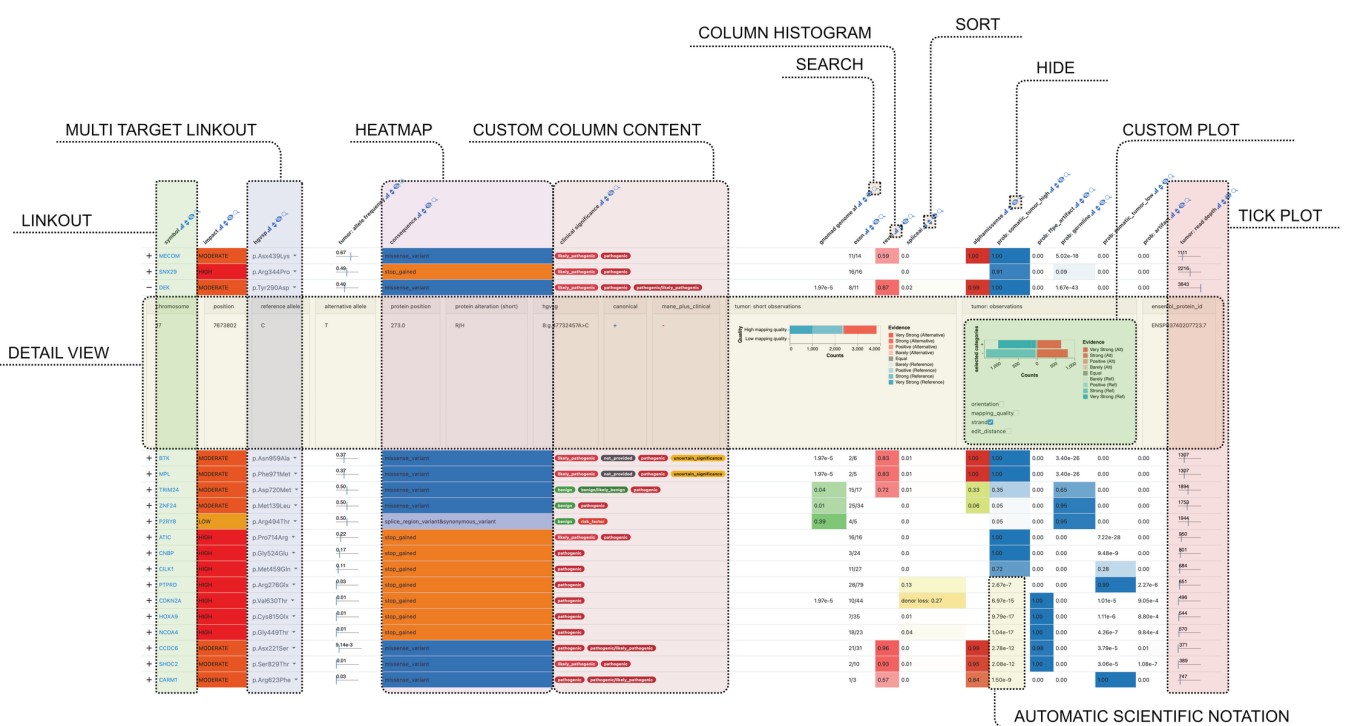

**Fig 2. Screenshot of a *Datavzrd* report with annotated visual elements and controls.** The underlying example dataset entails genomic variants along with various scores and predictions. Gene names and coordinates have been altered to de-identify the data. Source: https://github.com/snakemake-workflows/dna-seq-varlociraptor. An interactive version of this report is available under https://datavzrd.github.io/example-molecular-tumor-board.

repository using a simple command-line invocation. Afterwards it provides the user with a link to the repository settings where GitHub Pages [39] need to be enabled manually — once upon initial publication of the repository. This way authors can ensure transparency and continuous public availability of their results without the burden of maintaining a web server.

To further simplify the configuration process, *Datavzrd* also includes the `suggest` sub-command, which automatically generates a YAML configuration file based on one or more input CSV files. By inferring column types and suggesting suitable visualizations, it enables rapid report creation, allowing users to focus on their data rather than technical details.

The result is that Datavzrd's approach offers a more rapid path towards reproducible, transparent, comprehensive and informative communication of tabular scientific results that moreover is free of continued maintenance tasks (see Fig 1 and the Shiny vs. Datavzrd comparison in the supplement). Via compression methods and data partitioning strategies, *Datavzrd* is scalable towards big datasets without overwhelming the browser memory while still maintaining interactive capabilities. An overview of Datavzrd's out of the box visual interface capabilities can be found in Fig 2.

## Community driven extensibility

*Datavzrd* offers the ability to share recurrent configuration snippets for columns or views as so-called *spells*. This can be helpful since data analyses are likely to share columns with values that can be interpreted and visualized similarly, even when their field of application might be completely different. Examples for such cases are columns with boolean values (e.g. 0/1 or true/false) or columns with p-values or posterior probabilities.

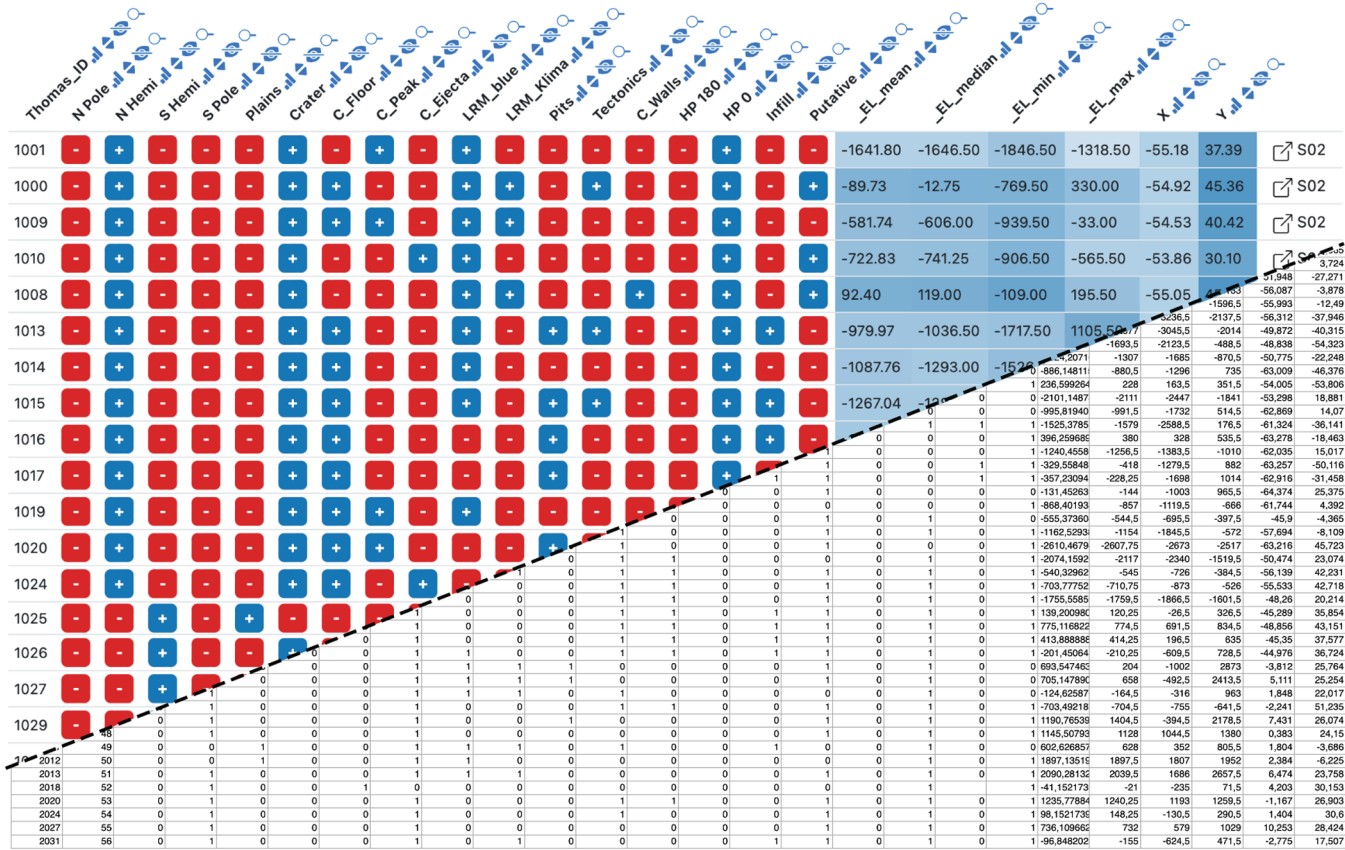

**Fig 3. Example Datavzrd view of an astronomy spreadsheet supplement of Barraud et al. [22].** Boolean columns have been rendered with Datavzrd's boolean spell [56], numeric columns have been rendered as heatmaps. The squarish link buttons at the far right of each row (generated by Datavzrd's dataset linking functionality, see section Interactivity and visuals) allow to jump to corresponding rows in other views.

A *spell* is a reusable template that defines the rendering of a specific column or view in a *Datavzrd* report. Each *spell* is written in YAML and supports parameterization through YTE [40], a YAML template engine, allowing users to pass custom values to the spell using the keyword 'with'. For example, a p-value spell might define a heatmap to visually represent statistical significance, where a configurable significance threshold determines the color gradient. In practice, users can apply a spell by specifying it directly within a report configuration as shown in the configuration of Fig 4.

*Spells* can be published in a central repository [41] and explored via an automatically updated catalog [42] so that others can reuse them in their own *Datavzrd* reports. This collaborative approach fosters a shared library of visualization techniques that grows and adapts to evolving research needs across disciplines, thereby obviating the need to reinvent the wheel for visualizing common data types. Since *spells* can also encode full views, they can be used to easily create reports with pre-defined visualizations for virtually any tool creating tabular output.

## Example reports for various scientific disciplines

To highlight Datavzrd's field-agnostic application across various scientific disciplines we created *Datavzrd* reports from the supplementary data of four recently published papers

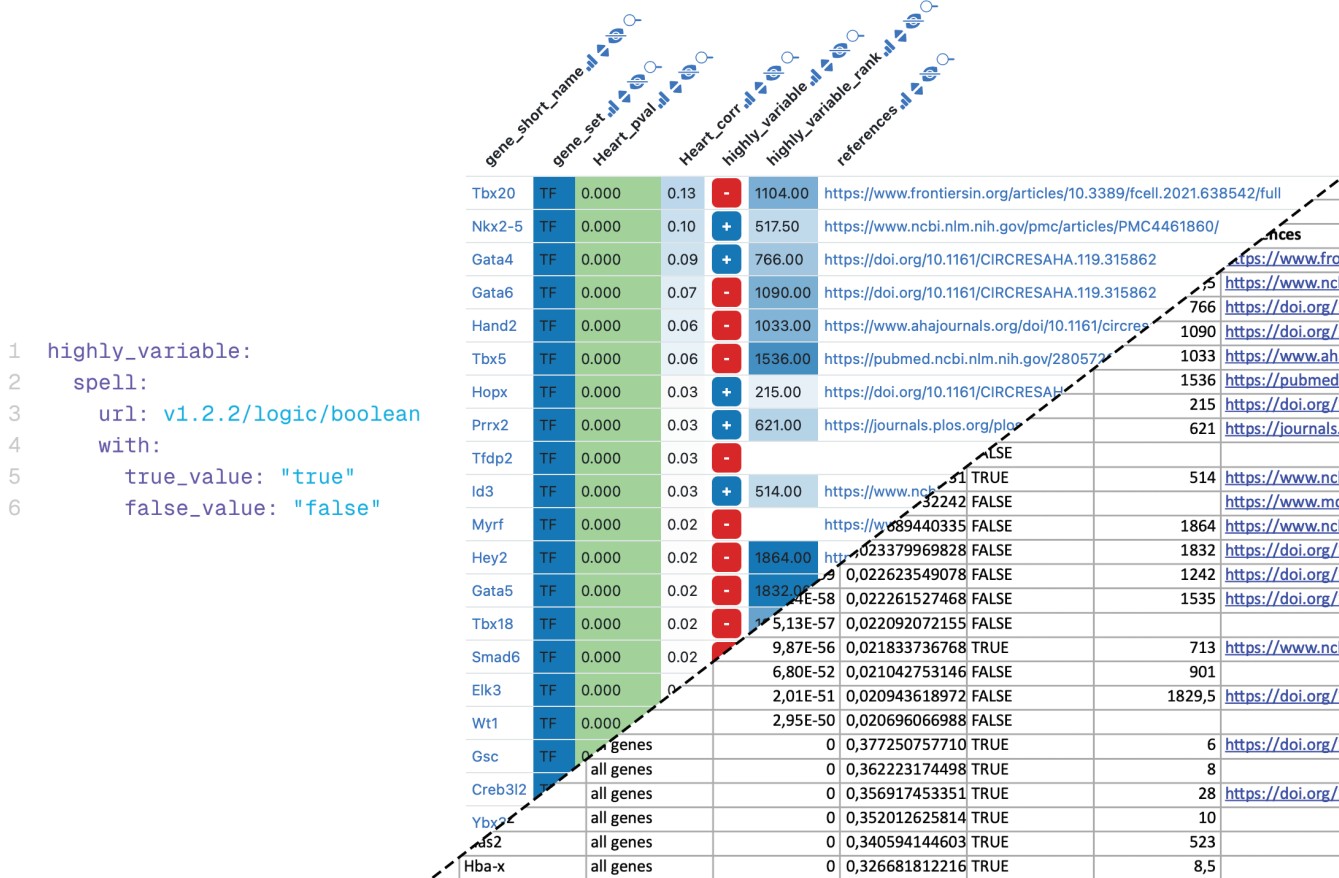

**Fig 4. Datavzrd boolean spell and its application to a gene table. Left:** Datavzrd spell for displaying boolean values of the column `highly_variable` of the report to the right. The `true_value` and `false_value` parameters define which values are rendered into a plus sign and a minus sign. **Right:** Example Datavzrd view of a bioinformatics spreadsheet supplement of Klein et al. [20]. Gene names have been rendered as links to a public database, boolean values have been rendered using Datavzrd's boolean spell [56] (see Community driven extensibility), other columns have been rendered as heatmaps with categorical or linear color scale.

[19–22], each originating from different fields: bioinformatics [43], social science [44], astronomy [45], and anthropology [46]. We have published the reports on GitHub pages using the `publish` subcommand outlined above. Tables 13 - 18 from Klein et al. [20] let the user navigate between driver genes of different cell types while also directly providing access to aforementioned genes in the Ensembl database. Table 25 of the same publication directly provides access to given genomic regions in the integrative genomics viewer [23] using a link-out. Instead of simply displaying numeric values with a given uncertainty (e.g. $-20.2 \pm 0.2$) as plain text, a table containing radiocarbon data from L. G. Sanjuán et al. [19] shows them in a convenient tick plot with the range of the uncertainty depicted as a red range bar and the central value represented as a blue tick. Individual views are presented exemplarily in Figs 3, 4, 5 and 6.

## Memory and storage footprint

We evaluated the storage utilization of *Datavzrd* reports in comparison to the raw input data and an Excel file. The analysis has been implemented as a fully reproducible Snakemake [5]

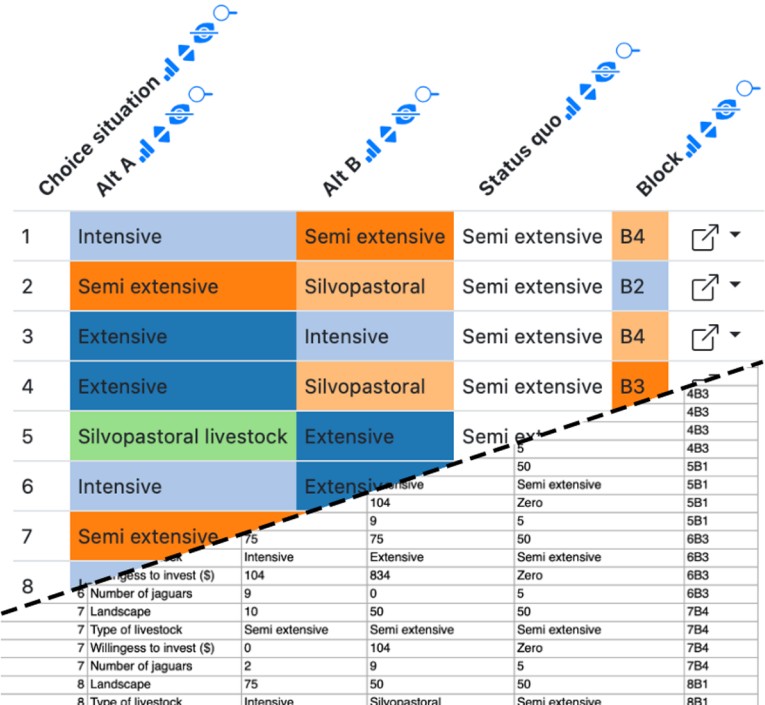

**Fig 5. Example Datavzrd view of a social science spreadsheet supplement of Alayón-Gamboa et al. [21].** Categorical columns have been rendered as heatmaps with a categorical color scale. The squarish link buttons at the far right of each row (generated by Datavzrd's dataset linking functionality, see section Interactivity and visuals) allow to jump to corresponding rows in other views.

workflow [47]. Our dataset [48] used for testing comprises 17 columns and 173534 rows with information about registered electric vehicles in the state of Washington. Seven of the columns contained numeric, ten contained nominal values.

In Fig 7, we compare the on disk storage footprint of two *Datavzrd* reports (one that represents the dataset entirely in memory and one that uses data partitioning for the same dataset) with the footprint of the CSV and Excel representation of the same dataset. One can see that the *Datavzrd* in-memory report occupies nearly as little space as the plain Excel file, while being (naturally) much smaller than the plain text CSV representation. Further, it can be seen that a bit less than half of the size is occupied by static resources like Javascript libraries and HTML pages. In contrast, the partitioned version of the report requires additional overhead for storing search indices and suffers from decreased compression rates, because each table page has to be compressed separately.

While it is apparent that both Excel and *Datavzrd* are able to successfully compress the dataset, Fig 7 indicates that compression rates of *Datavzrd* depend on the amount of table rows that can be compressed together (i.e. the page size or whether the entire dataset is held in memory). Fig 8 shows that the compression rate for the in-memory mode of *Datavzrd* exceeds that of Excel, while smaller page sizes lead to a monotone decrease of compression rates.

## Methods

We shed light on the central technical challenges and solutions within *Datavzrd* by considering portability, interactivity and scalability.

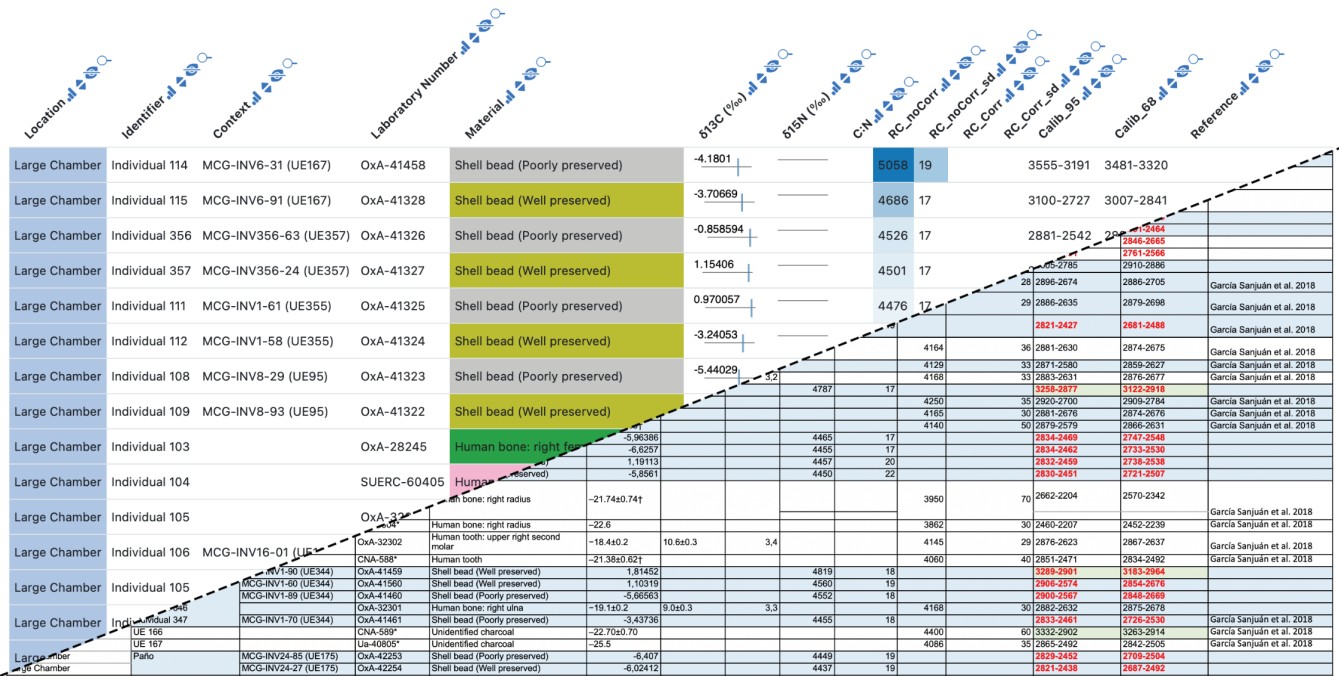

**Fig 6. Example Datavzrd view of an anthropology spreadsheet supplement of L. G. Sanjuán et al. [19].** Numerical values have been rendered as tick plots or heatmaps. Categorical columns have been rendered as heatmaps with categorical color scale.

## Portability

In the context of scientific research, seamless data exchange among researchers is a key factor for collaborative progress. With this in mind, *Datavzrd* implements various features targeting a high portability of the resulting reports.

A key feature of *Datavzrd* is a self-contained report architecture — eliminating the need for a dedicated backend or server. The entire report, structured as HTML files in a regular folder, is platform-agnostic, offering compatibility across all operating systems.

Browsers subject HTML pages to the same-origin-policy [8], which prohibits them from dynamically accessing resources hosted on domains other than their own. In case of standalone HTML pages that are accessed from the file system instead of being provided by a web server, this implies that the page may only access local files via static HTML tags. Hence it becomes impossible to dynamically load data stored in binary formats such as HDF5 [49], Parquet [37], or Flatbuffers [51], as well as to directly access databases like SQLite [52] or DuckDB [50]. To work around this limitation, *Datavzrd* stores data in JavaScript files, which are then loaded via static script tags. This approach requires regenerating the report whenever the input dataset changes; however, because data and configuration are completely separated, no updates to the latter are needed as long as the structure of the data remains unchanged. We recommend embedding the invocation of *Datavzrd* into automated workflows (e.g., using Snakemake [5], Nextflow [24], or CWL [25]) to facilitate reproducible report generation after data updates.

In addition to sharing full reports, *Datavzrd* provides a share button for each row, enabling to share individual rows of a dataset with others. This is solved by encoding the row data

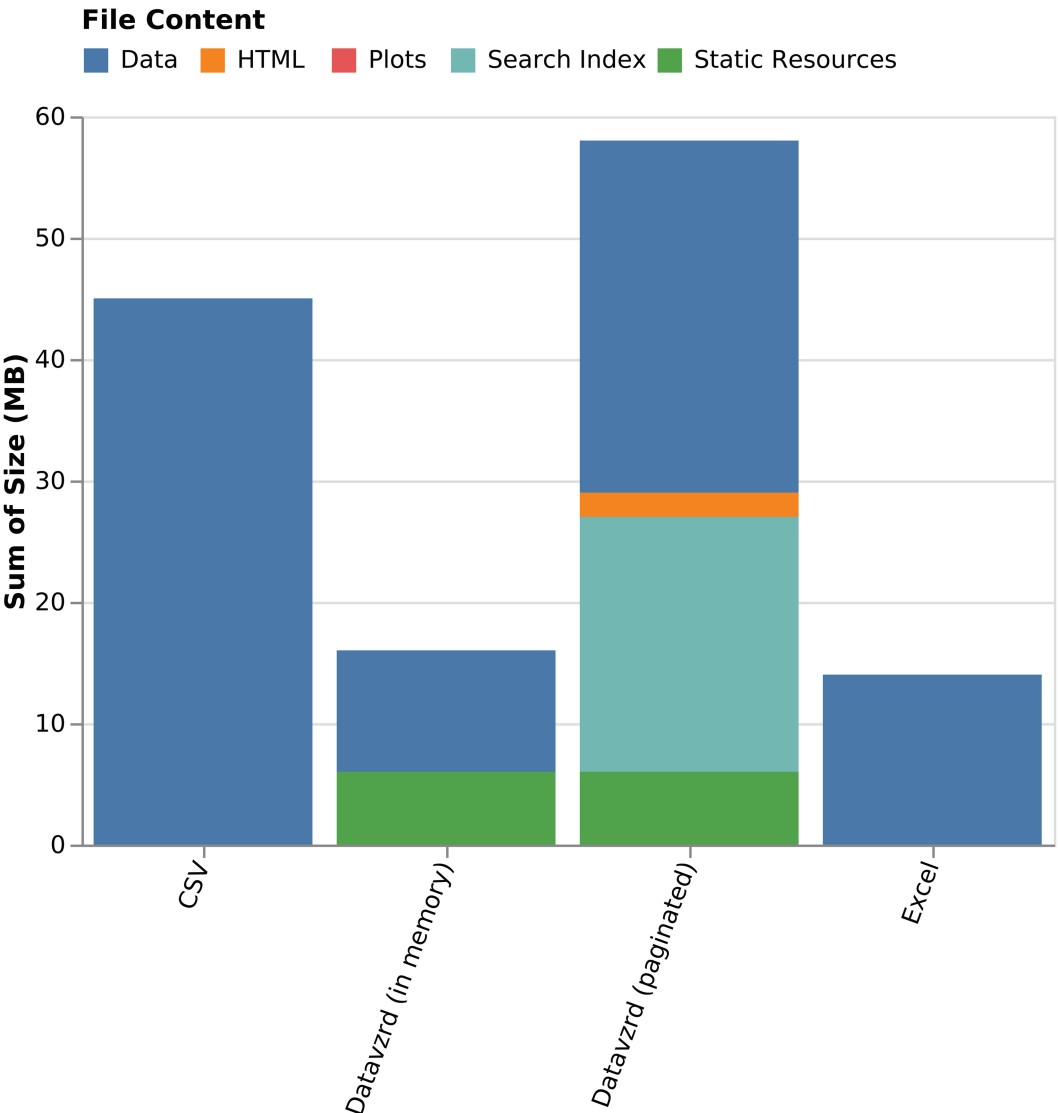

**Fig 7. Storage usage of *Datavzrd* reports compared with Excel and raw input data.**

into a URL parameter. The URL directs the user to a website hosted on GitHub that can display the exchanged dataset row by reading the URL parameter. Since the row data is in theory discoverable from URL parameters that might occur in server logs, the base URL of the HTML page can be customized through the configuration file, allowing organizations to ensure that sensitive data remains within their local network by hosting the previously mentioned HTML page themselves. *Datavzrd* can also encode this URL into a QR code, providing a practical method for presenters to share data during presentations.

Individual pages of the report can be exported into SVG format, for example in order to embed them into manuscript figures. Additionally, *Datavzrd* reports offer an export to Excel (XLSX). This way, they can serve as replacement for sharing spreadsheets without hampering the ability of collaborators to further analyze aspects of the data in their traditional spreadsheet environment.

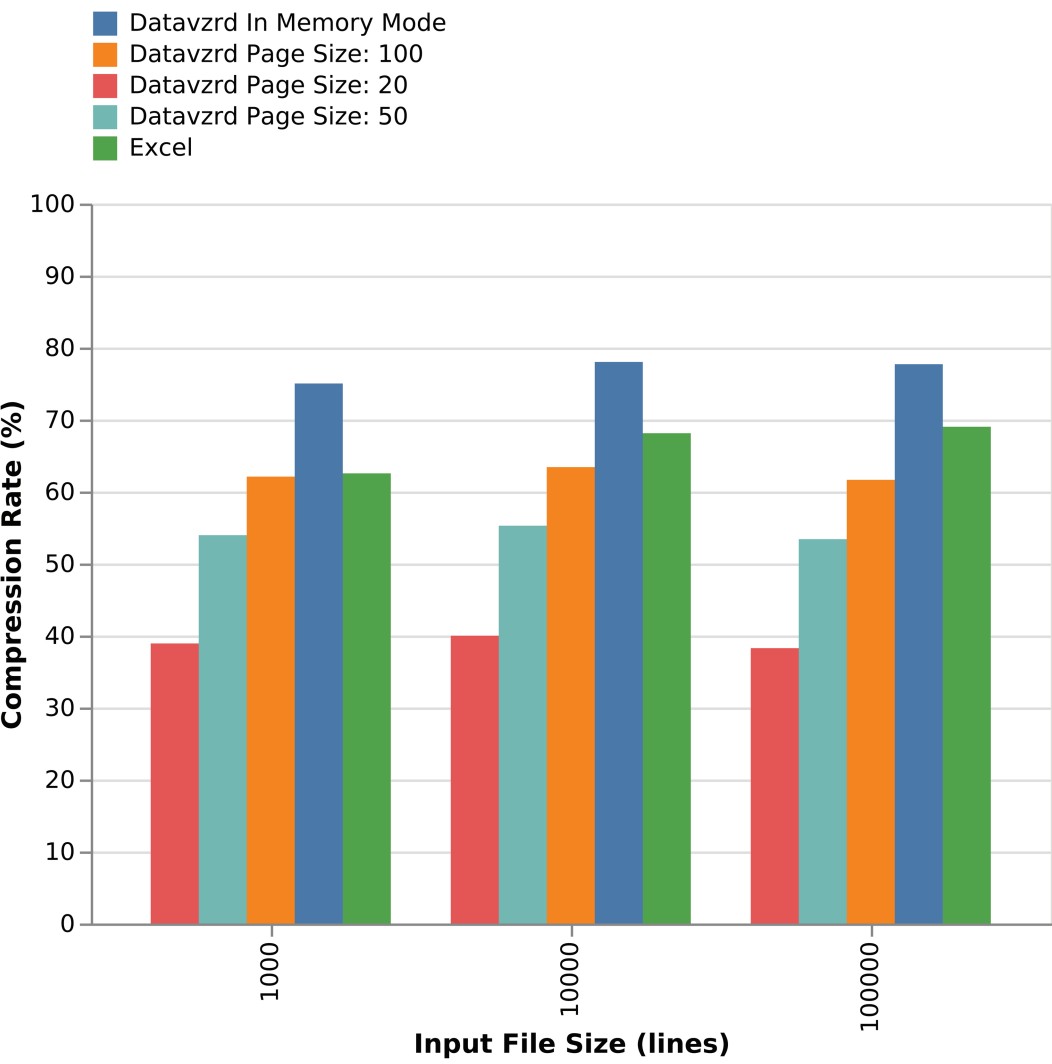

**Fig 8. Comparison of compression rates for increasing input sizes and different page sizes.**

## Interactivity and visuals

The most obvious kind of interaction with a table is to filter or search for rows of interest. Depending on whether the dataset is small enough to reside entirely in memory or not, *Datavzrd* offers either a filter or a search mode. The filter mode presents a widget for each column, which, depending on the type of data in the column, either allows to choose from discrete values, specify a search keyword, or select a numeric range. The widgets of the filter mode, along with the search mode, are displayed in Fig 9. Filters for different columns can be combined, yielding the intersection of the resulting rows. For a synthetic dataset with 100,000 rows and 30 columns (including nominal, integer, and string-valued columns), interactive filtering operations consistently complete in under 100 ms. Measurements were taken using the browser's `performance.now()` API in Safari 18.5 on macOS 13.7.6 with a 3.8 GHz Quad-Core Intel Core i5 and 48 GB of RAM. The search mode (due to constraints in the possibilities of representing large datasets out of memory in standalone HTML files) offers the ability

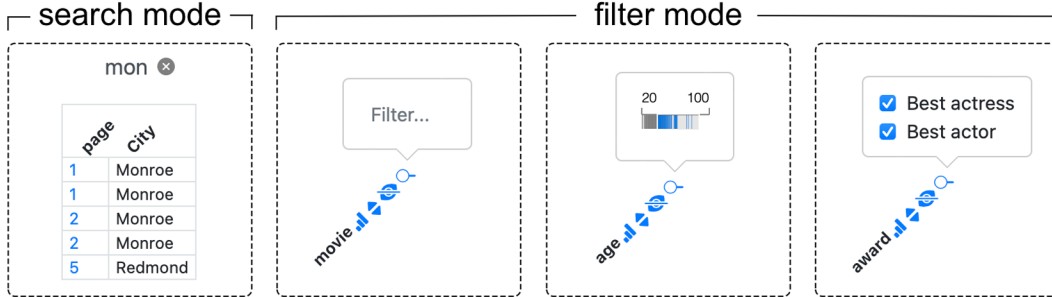

**Fig 9. Comparison of search and filter mode (text input, filter brush or multi-select).**

```
1   datasets:
2     table-a:
3       path: "table-a.csv"
4       links:
5         gene expression:
6           column: gene-name
7           table-row: table-b/gene
8         gene details:
9           column: gene-name
10          view: "gene-{value}"
```

**Fig 10. A *Datavzrd* dataset definition including different linkouts.** In this example, `table-a` creates a link to `table-b` where column `gene-name` of `table-a` and column `gene` of `table-b` match in value. With the `gene details` definition, `table-a` is linked to one of many tables existing for each value of `gene-name`.

to search for keywords or numbers and subsequently jump to the corresponding pages of the table, see section Scalability.

In practice, data can often be best represented by multiple tabular datasets that are related to each other. It may even be beneficial to divide wider tables into multiple ones, each containing a specific subset of columns [10]. Therefore, *Datavzrd* offers the capability to link between datasets, allowing the user to jump back and forth between corresponding rows of related tables.

The configuration of the datasets allows for the values of any column to be used as *foreign keys* to create a link to a row of another dataset with one column configured to be used as the *primary key*. Each *primary key* value must only occur once in the linked dataset to preserve uniqueness of the key lookup.

The same mechanism enables hierarchical linking by utilizing values from one column as foreign keys to connect with different tables. This feature allows for concise overview tables to establish connections with more detailed subtables, enhancing the exploration of specific details within the given data without requiring the entire hierarchical structure to be present in browser memory. An example for a dataset definition using both linking options is provided in Fig 10.

Columns in *Datavzrd* reports can be displayed in different modes: `normal`, `detail`, `available`, `pinned`, and `hidden`. These modes are designed to address the challenge

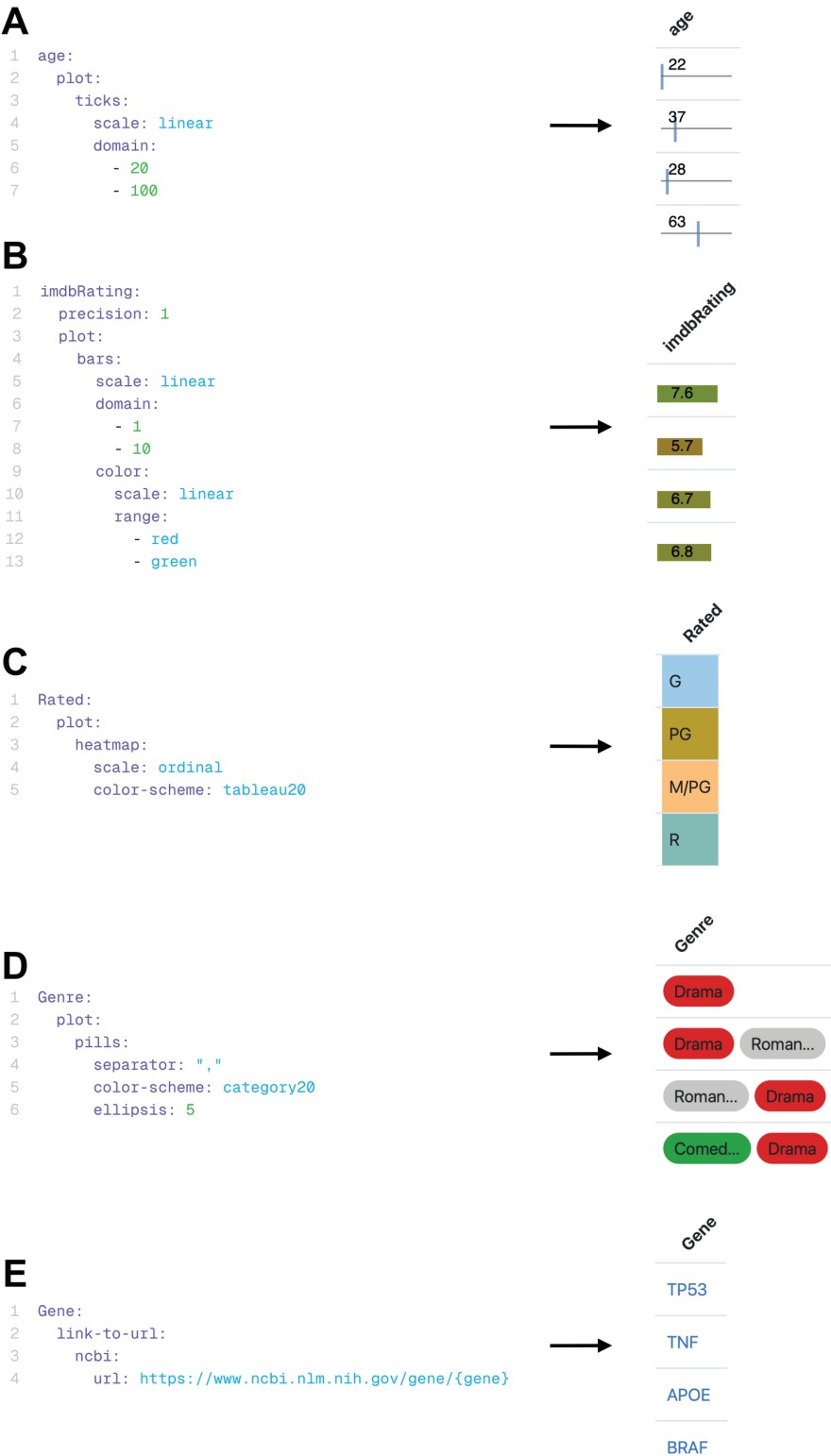

**Fig 11. Different pre-defined column visualizations of *Datavzrd*. Left**: corresponding YAML specification in the configuration file. **Right**: resulting column visualization. **A**: A *Datavzrd* tick plot definition with user-defined domain. **B**: A *Datavzrd* bar plot definition with user-defined domain and additional color domain. **C**: A *Datavzrd* heatmap for a column named Rated. **D**: Pill plot definition for a cell containing multiple values separated by any delimiter. **E**: YAML specification of a linkout to the NCBI gene database.

of maintaining clarity while accommodating a large amount of data. Research indicates that an ideal table typically contains 3-5 columns to ensure readability and prevent information overload [11]. While the exact number of columns will depend on their information content and the context of the table (on paper or on screen), *Datavzrd's* display modes offer the ability to follow these guidelines easily. In particular, the `hidden` mode allows for complete concealment of less critical columns, thereby keeping the table concise and focused. Meanwhile, the `detail` mode enables selective unfolding of additional columns per row as needed, allowing users to access more comprehensive data without compromising the table's clarity. Using `available` hides a column by default while still providing the user with the option to reveal it using a multi-select column interface on the top right when exploring the dataset. Complementary to this, `pinned` keeps the column visible in the table but excludes it from the interface, preventing it from being accidentally hidden and reducing clutter in the selection interface.

Guidelines for effective tables find that column headers should be kept short or use abbreviations [12]. This is possible using the `label` keyword allowing to specify column labels for the resulting report without the need to modify the given input dataset. To make tables self-explanatory, the `description` keyword can be used to provide additional information and context [12]. This keyword supports Markdown-formatted text [13] as well as LaTeX-style math equations, enabling the report creator to include detailed and complex annotations, ensuring the table can be understood by any observer. Additionally, *Datavzrd* provides the option to set a character limit for a column using the `ellipsis` keyword. This only shows the first *n* characters in the cell with the rest still accessible via a tooltip that appears while hovering the cell. External resources like knowledge databases can provide additional context with even more details to researchers. For easy access to external resources, *Datavzrd* provides the option to configure one or more linkouts for a cell using the value of the cell or other values of the same row to format the URL.

Datavzrd relies on Vega-Lite as its underlying engine for generating a wide variety of plots [3]. Internally, Vega-Lite specifications compile to Vega, which itself builds on D3.js for rendering. Datavzrd provides an even higher level of abstraction with its available pre-made plot configurations for `ticks`, `bars`, `pills` and `heatmaps`. When specified for a column (cf. Fig 11), these visualizations are automatically applied and configured to every cell within that column. Extending the existing plot configurations, *Datavzrd* also allows to include Vega-Lite plots that can be fully customized through JSON specifications, allowing researchers the flexibility to create visuals tailored to their specific needs and preferences. Through this mechanism, users can harness the full expressiveness of Vega-Lite, which supports a wide range of visualizations such as scatter plots, line and area charts, bar charts, heatmaps, geographic maps and faceted layouts (see examples for the extensive set of supported types). If neither of the previously mentioned options is suitable for a given task, configurations of columns may include any javascript function to process data and return plain HTML that is rendered into the respective table cell.

By using Vega-Lite plots, users can not only specify custom plots using values of a single row to display inside a table but also harness the entire dataset to craft plots that convey meaningful insights independently of the tabular view and stand as a single view inside the report. Full plot views in *Datavzrd* retain the previously introduced linking feature for datasets, enabling users to click on data points within a plot for navigating to corresponding rows of complementary tables.

Finally, beyond tabular and plot views, *Datavzrd* offers the possibility to configure plain HTML views, allowing to embed any kind of custom interface or visualization.

## Scalability

For ensuring scalability towards big datasets, *Datavzrd* uses a combination of different compression techniques. Initially, the data is compressed using JSONM [53]. JSONM uses memoization principles on JSON objects in order to eliminate any unnecessary repetition of data. In particular, it stores keys only once, such that objects of the same kind can be represented via arrays of values. The JSONM representation is then compressed with lz-string [54], a JavaScript library that implements Lempel-Ziv compression algorithms while ensuring that the resulting compressed word can be represented as a Javascript string (thus not containing any invalid binary only characters).

Any tabular dataset encoded by the above approach will, upon rendering the report's HTML pages, still be represented in uncompressed form in the browser memory. If the uncompressed dataset size exceeds the available system memory this can lead to slowdowns or crashes. In order to avoid this situation, *Datavzrd* partitions large datasets into multiple chunks, each of them being only loaded when the corresponding table page in the report is accessed. The threshold for and size of these chunks can be configured upon report generation (default: 20000 rows). Since those chunks cannot be loaded dynamically due to the same-origin-policy, we encode each table page as a separate HTML file. Page sizes may be configured by the user to address variations in performance across different machines and account for datasets with differing numbers of columns.

Searching throughout such chunked datasets, paginated across multiple HTML pages, presents a challenge as direct access to these pages is restricted by the same-origin-policy. To overcome this limitation, *Datavzrd* implements a pre-built search index for each column, providing an effective solution for search and navigation across the distributed dataset. The index includes the values of each column alongside their respective page locations, and is displayed upon request using an embedded iframe. This way, we avoid that the indexes of all columns are loaded entirely into memory when rendering a page. Within the index table, the user can filter for values of interest, such that the pages of occurrence are displayed and can be navigated to. Upon the latter, the row corresponding to the value of interest is highlighted.

If the dataset is small enough to fit entirely into memory, *Datavzrd* can instead rely on a versatile filtering mechanism which enables to dynamically select and combine data ranges across multiple columns.

## Conclusion

Tabular data, often scattered across multiple tables, is the primary output of data analyses in virtually all scientific fields. Exchange and communication of tabular data is therefore a central challenge in data science. So far, this has been mostly handled by spreadsheet applications like Microsoft Excel - offering limited visual and interactive capabilities as well as potential problems with compatibility, replicability and value encoding [4] - or server based approaches like R Shiny, Lumen or dedicated web applications, requiring maintenance of web servers and potentially extensive imperative programming while limiting sustainability and reproducibility upon publication.

To overcome this situation, we have developed *Datavzrd*, which unifies rich visual capabilities with the portability offered by plain spreadsheets. It enables the generation of server-free, portable, interactive visual reports on tabular data that can be rapidly configured via a declarative specification language, in many cases obviating the need for imperative programming. Once a configuration file is created, *Datavzrd* can be repeatedly applied to various datasets of the same structure, offering automated, efficient, and scalable table visualization. Linking features, various display modes, extensive visualization capabilities, and the ability to specify

fully custom non-tabular views, simplify structured dataset exploration, even across multiple related tables. *Datavzrd's* versatile export functionalities offer straightforward inclusion into manuscript figures. Via *spells*, custom column or view configurations can be shared and developed as a community project.

Future work on *Datavzrd* will entail a further improvement of the data compression and decompression strategy. One approach is to directly compress binary encoded JSONM with lz-string. Given that a machine independent decoding of the binary encoding from within Javascript can be realized, this approach would provide a substantial further reduction of the space required for numeric values. Furthermore, we plan to extend support to additional data types, such as temporal data. Currently, *Datavzrd* can already be integrated into other standalone reporting tools like for example Snakemake reports. Using web components [55], we plan to generalize the user interface so that its components can be embedded into arbitrary web application frameworks. Currently, *Datavzrd's* aggregation capabilities are limited to simple, per-column summaries such as histograms. More complex operations like window functions or pivot tables are not supported. We see this as a deliberate design choice: such aggregations are better handled upstream in the analysis workflow [5,24,25], where they can be executed transparently and reproducibly. In future versions, we plan to extend Datavzrd with lightweight interactive features such as scatterplots for user-selected column pairs.

In summary, *Datavzrd* streamlines, simplifies, and enriches the exchange and communication of tabular data, one of the most abundant outputs of scientific data analyses, while providing seamless scaling from small tables to thousands or even millions of rows without losing the ability of interactive and visually rich exploration. All these capabilities can be used on any kind of tabular data, and the resulting reports can be used as supplementary files for published manuscripts, or shared with collaborating researchers before publication. This enables the collaborators and readers to interactively explore the data underlying the findings of an analysis or published manuscript directly in their web browser without imposing additional maintenance or implementation burdens to the analysis authors. Likewise, it enables journals or scientific data hosting services like Zenodo [1] to sustainably ensure the availability of also the interactive resources associated with a manuscript by simply hosting a set of static HTML files instead of requiring the maintenance of server processes.

## Software availability

*Datavzrd* is implemented as a command line application with the *Rust* [32] programming language. It is available as an MIT licensed open source software via *Github* [33], can be installed via *Cargo* [34] and *Conda* [35] or used as a *Snakemake* wrapper [36] for rapid integration into reproducible data analysis workflows.

## Supporting information

**S1 Appendix. Comparison of Shiny and Datavzrd for Interactive Table Creation.** We illustrate how Datavzrd supports rich, per-cell visualizations using Vega-Lite and compare its concise, declarative configuration against a minimal Shiny implementation. Example configurations, rendered outputs, and setup instructions are provided for both tools.
(PDF)

**S2 File. Example report for Fig 2.** Interactive Datavzrd report showcasing genomic variants with associated scores and predictions in a molecular tumor board context. The dataset has been de-identified by altering gene names and coordinates; see the original workflow at

https://github.com/snakemake-workflows/dna-seq-varlociraptor and explore the interactive report at https://datavzrd.github.io/example-molecular-tumor-board.
(ZIP)

## Acknowledgments

We thank everyone who contributed by opening issues or submitting pull requests to *Datavzrd*. Although not listed as co-authors, their input helped refine the software and is acknowledged with appreciation.

## Author contributions

**Conceptualization:** Felix Christian Wiegand, Johannes Köster.

**Methodology:** Felix Christian Wiegand, Johannes Köster.

**Software:** Felix Christian Wiegand, Johannes Köster.

**Supervision:** Johannes Köster.

**Writing – original draft:** Felix Christian Wiegand, Johannes Köster.

**Writing – review & editing:** Felix Christian Wiegand, David Lähnemann, Felix Mölder, Hamdiye Uzuner, Adrian Prinz, Alexander Schramm, Johannes Köster.

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
