## [Decision Letter · Decision Letter 0]

28 May 2025

PONE-D-25-17640Datavzrd: Rapid programming- and maintenance-free interactive visualization and communication of tabular dataPLOS ONE

Dear Dr. Wiegand,

Thank you for submitting your manuscript to PLOS ONE. After careful consideration, we feel that it has merit but does not fully meet PLOS ONE’s publication criteria as it currently stands. Therefore, we invite you to submit a revised version of the manuscript that addresses the points raised during the review process.

We look forward to receiving your revised manuscript.

Kind regards,

Vivek Kumar, Ph. D.

Academic Editor

PLOS ONE

Journal Requirements:

2. Please note that your Data Availability Statement is currently missing the repository name and/or the DOI/accession number of each dataset OR a direct link to access each database. If your manuscript is accepted for publication, you will be asked to provide these details on a very short timeline. We therefore suggest that you provide this information now, though we will not hold up the peer review process if you are unable.

4. We are unable to open your Supporting Information file “datavzrd-report-supplement.zip”. Please kindly revise as necessary and re-upload

Additional Editor Comments:

The paper introduces Datavzrd, a declarative framework for generating server-free, interactive HTML visualizations from tabular data. It is clearly written and presents an appealing tool that could benefit a wide range of disciplines working with structured data. The framework emphasizes ease of use, portability, and client-side rendering, reducing the need for complex server infrastructure or imperative programming.

However, the manuscript lacks detailed workflows, comprehensive comparisons with existing tools (e.g., D3, Plotly, Dash), and quantified performance evaluations. Technical details, including supported data formats and visualization configurations, are either missing or insufficiently explained. Visual elements like figures and code snippets also need improvement for clarity and completeness.

The authors should ensure that the supporting files (such as datavzrd-supplement.pdf and datavzrd-report-supplement.zip and its contents) are accessible in the manuscript’s Supporting Information section so they are not missed by the reviewers and are eventually accessible to the readers of the publication as well.

Overall, Datavzrd shows good potential, but the paper would benefit from additional empirical support, clearer technical exposition, and a more balanced discussion of its limitations and place within the broader visualization tool landscape.

Reviewers' comments:

Reviewer's Responses to Questions

**Comments to the Author**

1. Is the manuscript technically sound, and do the data support the conclusions?

Reviewer #1: Partly

Reviewer #2: Yes

2. Has the statistical analysis been performed appropriately and rigorously? 

Reviewer #1: N/A

Reviewer #2: N/A

3. Have the authors made all data underlying the findings in their manuscript fully available?

Reviewer #1: Yes

Reviewer #2: Yes

4. Is the manuscript presented in an intelligible fashion and written in standard English?

Reviewer #1: Yes

Reviewer #2: Yes

5. Review Comments to the Author

Reviewer #1: The paper presents a framework called Datavzrd that allows easy visualization of tabular data via HTML pages. It does not report substantial research findings, but presents a new research method which could be interesting for many disciplines dealing with tabular data.

The paper is written in comprehensible language and presented in an appealing style. It claims many advantages of their tool over existing alternatives, but misses evidence for these claims.

Major comments:

1. The authors claim that their tool allows more efficient visualizations with less code than existing alternatives, in particular Shiny. This is argued in text, but not substantiated by examples. An example workflow for a simple data table should be included, which puts the Datavzrd approach in contrast to Shiny (similar to the supplementary material plus brief explanations).

2. In Sec. 1 Excel and the xlsx format are mentioned. It is not clear, however, which other – esp. non-proprietary – file formats are supported and how the tool is applied to these.

3. Furthermore it remains unclear how the Datavzrd definition of visualizations interacts with the dataset. There are very few code snippets (e.g. Fig. 4) which, however, do not show how visualizations are defined. An example workflow as suggested in comment 1 could help.

Minor comments:

4. In the related works section of the introduction, D3, Plotly, and Dash could be mentioned as further widely used tools (also requiring lots of code, though).

5. Figure 2 is too small and thus unreadable. It should be rotated by 90 degrees to fill a whole page.

6. Code and visualization in Fig. 4 do not match. Furthermore, the very small code snippet does not explain what a "spell" is and how the values true and false are translated in visual representations.

7. Several references are missing bibliographical information such as journal/conference, page numbers etc., such as [14], [16], [17], [22]. Furthermore, journal names should be capitalized. More care should be taken when preparing the next version.

Reviewer #2: In this paper, the authors introduce Datavzrd, a framework designed for generating server-free, portable, and interactive visual reports from tabular data. The system leverages a declarative specification language to facilitate rapid report configuration, eliminating the need for imperative programming. The paper is generally well-written and is supported by illustrative examples and clear explanations.

One notable advantage the authors might emphasize is that client-side JavaScript frameworks reduce server load, as all rendering, interactivity, and event handling occur in the client’s browser. Given the capabilities of modern PCs and smartphones, this approach is both practical and scalable.

However, to substantiate the authors' claim regarding the uniqueness of Datavzrd, it is important to include a more comprehensive comparison with existing client-side JavaScript frameworks. Specifically, a discussion of alternatives such as D3.js, Vega, Plotly.js, and even older technologies like XSLT would strengthen the paper by placing Datavzrd within the context of existing tools.

Many server-based reporting tools provide advanced aggregation capabilities such as window functions, multi-level aggregations, predictive analytics, and pivot operations. These typically require multiple data passes and significant memory. The authors should clarify the scope of Datavzrd’s aggregation capabilities, which currently seem limited to simple aggregations, such as column histograms and heatmaps.

On page 5, the authors state: “Via compression methods and data partitioning strategies, Datavzrd is scalable towards big datasets without overwhelming the browser memory while still maintaining interactive capabilities.” The authors should quantify performance characteristics to improve clarity and reproducibility. Consider including details such as:

Performance benchmarks: e.g., “Interactive operations such as filtering respond in under 300 milliseconds with datasets up to 5 GB in size.”

Test environment: e.g., “Tested on a machine with 32 GB RAM, Intel i7 CPU, running Windows 11.”

Visualization types supported: e.g., “Line chart, heatmap, 3D surface plot.”

Resource usage: e.g., “CPU and RAM utilization during peak operations.”

Avoid vague claims such as “handles big files efficiently” without accompanying empirical evidence.

Additional Comments:

Page 4:

Figure 2 is too small to be legible. The authors should enlarge the figure to ensure that the visualizations are clearly discernible.

Page 4:

The manuscript states:

“Datavzrd provides a publish subcommand, which automates the upload of reports to a newly created or existing GitHub repository and enables hosting via GitHub Pages with a single click.”

The phrase "with a single click" is misleading. A more accurate description would be:

“...and enables hosting via GitHub Pages using a simple command-line invocation.”

The process requires executing a command in the following format:

datavzrd publish --repo-name <repo_name> --report-path <report_path> [--org <organization>]

This is not a one-click operation and should be described as such.

Page 10:

The authors note:

“Datavzrd stores data in JavaScript files, which are then loaded via static script tags.”

This is an important implementation detail. However, the authors should also acknowledge that when data updates are needed, the visualizations must be regenerated using the Datavzrd command line. This introduces some operational overhead, and its mention would provide a more balanced and transparent evaluation.

Page 14:

The statement:

“For the first time, this enables the collaborators and readers to interactively explore the data underlying the findings…” is overly assertive. Without rigorous comparative analysis, claims of precedence should be avoided.

Evaluation of the Linked Resources:

Live Demo

https://datavzrd.github.io/example-report-moscot/Suppl.%20Tab.%2013/index_1.html

The Birth date field is interpreted as a string, preventing effective range-based searches. Additionally, the column histogram is not time-ordered. These are areas where usability could be improved.

Demonstration Table

https://datavzrd.github.io/example-report-moscot/Suppl.%20Tab.%207/index_1.html

In the third field, after initiating a search, the data range in the histogram appears confusing. This could hinder interpretability and should be refined.

GitPod Tutorial

The tutorial is well-structured and user-friendly. However, a bug exists: after executing the show-report-url command and clicking the generated link, the hide button on top of each field does not actually hide the corresponding field in the report. This issue should be addressed.

GitHub Live Preview

https://datavzrd.github.io/datavzrd/index.html

The font size in the Markdown file is inconsistent and should be normalized for a professional appearance.

The hide button on top of each field does not actually hide the corresponding field in the following links:

https://datavzrd.github.io/datavzrd/oscars/index_1.html

https://datavzrd.github.io/datavzrd/movies/index_1.html

Additionally, data visualizations are missing in the following links:

https://datavzrd.github.io/datavzrd/movies-plot/index_1.html

https://datavzrd.github.io/datavzrd/oscar-plot/index_1.html</organization></report_path></repo_name>

6. PLOS authors have the option to publish the peer review history of their article (what does this mean?). If published, this will include your full peer review and any attached files.

Reviewer #1: No

Reviewer #2: **Yes: **Xiaorong Cao

---

## [Author Response · Author response to Decision Letter 1]

27 Jun 2025

We thank both reviewers for the comprehensive and positive reviews. In the following, we answer each comment in detail.

Reviewer 1

General comments The paper presents a framework called Datavzrd that allows easy visualization of tabular data via HTML pages. It does not report substantial research findings, but presents a new research method which could be interesting for many disciplines dealing with tabular data. The paper is written in comprehensible language and presented in an appealing style. It claims many advantages of their tool over existing alternatives, but misses evidence for these claims.

Response We thank the reviewer for the constructive summary and for recognizing the potential interdisciplinary relevance and clarity of the manuscript. We have re- worked various areas of the manuscript to better and more clearly support the claimed advantages. See our responses to specific comments below.

Comment 1 The authors claim that their tool allows more efficient visualizations with less code than existing alternatives, in particular Shiny. This is argued in text, but not substantiated by examples. An example workflow for a simple data table should be included, which puts the Datavzrd approach in contrast to Shiny (similar to the supplementary material plus brief explanations).

Response

We have extended the comparison in the supplement with explanations on the usage of both Shiny and Datavzrd. Additionally we have added another new supplementary section that further explains the comparison of the Shiny and Datavzrd configuration example. The supplement highlights that the Datavzrd approach requires about half of the lines compared to Shiny (28 vs. 49), involves no programming, and avoids data manipulation or plotting libraries. Instead, the full configuration is expressed in a short, human-readable YAML file. This addition should further substantiate our claim that Datavzrd enables more concise and accessible visualizations compared to imperative frameworks like Shiny.

Comment 2 In Sec. 1 Excel and the xlsx format are mentioned. It is not clear, however, which other – esp. non-proprietary – file formats are supported and how the tool is applied to these.

Response We have clarified the supported file formats in the revised manuscript. In short, Datavzrd supports CSV (and TSV) or Apache Parquet1 as input formats, both being non-proprietary. For exporting tabular views from the browser display, Datavzrd supports CSV and Excel.

Comment 3 Furthermore it remains unclear how the Datavzrd definition of visualiza- tions interacts with the dataset. There are very few code snippets (e.g. Fig. 4) which, however, do not show how visualizations are defined. An example workflow as suggested in comment 1 could help.

Response We have unified the examples of column configurations into a single figure and extended them with a bar and pill plot definition. Furthermore we expanded the supplement with an example of a custom plot definition that uses Vega-Lite syntax. In short, one can say that the user specifies a Datavzrd configuration file. When applying that file to one or more tables (by invoking Datavzrd as a command line tool), Datavzrd uses the specified visualizations to render the tables into self-contained interactive HTML which can then be send around, stored in some cloud, be attached as a supplementary to a manuscript, or hosted via static web servers (including e.g. github pages). To clarify the core workflow, we expanded the manuscript with an explanation on how configuration and dataset interact:

Using simple YAML-based declarative specifications, users define datasets along with the desired visualizations for each column. The self-contained report is then generated by exe- cuting Datavzrd with the command datavzrd path/to/config.yaml -o path/to/output.

1 https://parquet.apache.org/

2

Comment 4 In the related works section of the introduction, D3, Plotly, and Dash could be mentioned as further widely used tools (also requiring lots of code, though).

Response We thank the reviewer for the suggestion. In response, we have added a sentence to the related works section of the introduction mentioning Dash, D3.js, and Plotly, and positioning them in the broader spectrum of solutions:

Similar to Shiny, Dash allows to create interactive dashboards with Python code, but also requiring a running server process. [...] Of course, charting libraries such as D3.js or Plotly enable the creation of interactive visualizations, but using them within tables requires substantial bespoke coding and web-development effort.

Furthermore we have clarified the position of Vega-Lite, Vega and D3.js within Datavzrds ecosystem in the interactivity section:

Datavzrd relies on Vega-Lite as its underlying engine for generating a wide variety of plots. Internally, Vega-Lite specifications compile to Vega, which itself builds on D3.js for rendering. Datavzrd provides an even higher level of abstraction with its available pre-made plot configurations for ticks, bars, pills and heatmaps. When specified for a column (cf. Figure 9), these visualizations are automatically applied and configured to every cell within that column.

Comment 5 Figure 2 is too small and thus unreadable. It should be rotated by 90 degrees to fill a whole page.

Response We have rotated the figure accordingly to improve readability.

Comment 6 Code and visualization in Fig. 4 do not match. Furthermore, the very small code snippet does not explain what a ”spell” is and how the values true and false are translated in visual representations.

Response We fully agree with the reviewer. In response, we have switched Figs. 4 and 5 to better align the code with the corresponding visualization. We also improved the explanation of how the spell definition on the left corresponds to the rendered report example on the right.

Comment 7 Several references are missing bibliographical information such as jour- nal/conference, page numbers etc., such as [14], [16], [17], [22]. Furthermore, journal names should be capitalized. More care should be taken when preparing the next version.

Response We thank the reviewer for pointing out the missing details in some of the references. We have carefully reviewed all references in the manuscript and, to the best of our ability, added all available bibliographic information, including journal/conference, DOIs, and page numbers where applicable.

For the specific reference mentioned by the reviewer, we have updated the entry to include additional details. A summary of the changes for the specifically mentioned references is provided in the list below.

14 A multi-analytical study of [...] We have added more information including the article identifier eadp1917 since Science Advances doesn’t use traditional page numbers.

16 Sustainable data analysis with Snakemake We have added the missing DOI and issue number. F1000 does not provide page numbers.

17 The Not-So-Same-Origin Policy We have added a missing author as well as a link to article.

22 Zenodo We could not find a better option of citing Zenodo themselves. We use the BibTex entry they provide to cite them at the bottom of their ”About” page: https://about.zenodo.org.

Reviewer 2

General comments In this paper, the authors introduce Datavzrd, a framework de- signed for generating server-free, portable, and interactive visual reports from tabular data. The system leverages a declarative specification language to facilitate rapid report configuration, eliminating the need for imperative programming. The paper is generally well-written and is supported by illustrative examples and clear explanations.

Response We thank the reviewer for the positive and encouraging feedback. We are glad the manuscript provided clear explanations and illustrative examples that were helpful in conveying the core ideas and design of Datavzrd.

Comment 1 One notable advantage the authors might emphasize is that client-side JavaScript frameworks reduce server load, as all rendering, interactivity, and event han- dling occur in the client’s browser. Given the capabilities of modern PCs and smart- phones, this approach is both practical and scalable.

4

Response We thank the reviewer for this helpful suggestion. We have incorporated this point into the manuscript:

Additionally, it also eliminates server maintenance and reduces server load by shifting processing load to the browser (without hampering scalability, see section Scalability).

Comment 2 However, to substantiate the authors’ claim regarding the uniqueness of Datavzrd, it is important to include a more comprehensive comparison with existing client-side JavaScript frameworks. Specifically, a discussion of alternatives such as D3.js, Vega, Plotly.js, and even older technologies like XSLT would strengthen the paper by placing Datavzrd within the context of existing tools.

Response We have extended our discussion of Datavzrd’s functionality in the context of other approaches in the introduction (thereby also adding XSLT, Dash, D3.js, Vega, Plotly.js):

Naturally, any general purpose programming language (like e.g. Python or R) or trans- formation languages (e.g. XSLT) could be used (also e.g. in comparison with helper libraries like great-tables) to implement entirely custom reporting. [...] Similar to Shiny, Dash allows to create interactive dashboards with Python code, but also requiring a run- ning server process. [...] Of course, charting libraries such as D3.js or Plotly enable the creation of interactive visualizations, but using them within tables requires substantial bespoke coding and web-development effort.

Furthermore we have clarified the position of Vega-Lite, Vega and D3.js within Datavzrds ecosystem in the interactivity section:

Datavzrd relies on Vega-Lite as its underlying engine for generating a wide variety of plots. Internally, Vega-Lite specifications compile to Vega, which itself builds on D3.js for rendering. Datavzrd provides an even higher level of abstraction with its available pre-made plot configurations for ticks, bars, pills and heatmaps.

Comment 3 Many server-based reporting tools provide advanced aggregation capabil- ities such as window functions, multi-level aggregations, predictive analytics, and pivot operations. These typically require multiple data passes and significant memory. The authors should clarify the scope of Datavzrd’s aggregation capabilities, which currently seem limited to simple aggregations, such as column histograms and heatmaps.

Response We clarified the limit of Datavzrd’s aggregation capabilities together with an explanation on why these are not developed further at the moment:

Currently, Datavzrd’s aggregation capabilities are limited to simple, per-column sum- maries such as histograms. More complex operations like window functions or pivot tables are not supported. We see this as a deliberate design choice: such aggregations are better handled upstream in the analysis workflow, where they can be executed transpar- ently and reproducibly. In future versions, we plan to extend Datavzrd with lightweight interactive features such as scatterplots for user-selected column pairs.

Comment 4 On page 5, the authors state: “Via compression methods and data par- titioning strategies, Datavzrd is scalable towards big datasets without overwhelming the browser memory while still maintaining interactive capabilities.” The authors should quantify performance characteristics to improve clarity and reproducibility. Consider including details such as: Performance benchmarks: e.g., “Interactive operations such as filtering respond in under 300 milliseconds with datasets up to 5 GB in size.” Test environment: e.g., “Tested on a machine with 32 GB RAM, Intel i7 CPU, running Win- dows 11.” Visualization types supported: e.g., “Line chart, heatmap, 3D surface plot.” Resource usage: e.g., “CPU and RAM utilization during peak operations.” Avoid vague claims such as “handles big files efficiently” without accompanying empirical evidence.

Response We conducted additional benchmarking experiments on a large synthetic dataset and added the resulting quantitative performance characteristics for filter opera- tion times and the corresponding hardware and software specifications to the manuscript.

For a synthetic dataset with 100,000 rows and 30 columns (including nominal, integer, and string-valued columns), interactive filtering operations consistently complete in under 100 ms. Measurements were taken using the browser’s performance.now() API in Safari 18.5 on macOS 13.7.6 with a 3.8 GHz Quad-Core Intel Core i5 and 48 GB of RAM.

Regarding supported visualization types, we clarified that, in addition to Datavzrd’s predefined plot types (see Section “Feature overview” incl. new figure), users can embed fully custom Vega-Lite specifications. This enables the creation of a wide range of visualization types supported by Vega-Lite, including scatter plots, maps, faceted views, and more.

Through this mechanism, users can harness the full expressiveness of Vega-Lite, which supports a wide range of visualizations such as scatter plots, line and area charts, bar charts, heatmaps, geographic maps and faceted layouts (see https://vega.github. io/ vega-lite/ examples/ for the extensive set of supported types).

Comment 5 Figure 2 is too small to be legible. The authors should enlarge the figure to ensure that the visualizations are clearly discernible.

Response We have rotated the figure to increase the size and readability.

Comment 6 The manuscript states: “Datavzrd provides a publish subcommand, which automates the upload of reports to a newly created or existing GitHub repository and enables hosting via GitHub Pages with a single click. The phrase ”with a single click” is misleading. A more accurate description would be: “...and enables hosting via GitHub Pages using a simple command-line invocation.” The process requires executing a command in the following format: datavzrd publish --repo-name --report-path [--org ] This is not a one-click operation and should be described as such.

Response This has been indeed phrased very badly and is misleading. The click operation is needed afterwards to enable GitHub pages for the repository since this is not possible via the GitHub CLI. We have clarified the section of the manuscript accordingly:

For enhanced accessibility and ease of hosting, Datavzrd provides a publish subcom- mand, which automates the upload of reports to a newly created or existing GitHub repository using a simple command-line invocation. Afterwards it provides the user with a link to the repository settings where GitHub Pages need to be enabled manually — once upon initial publication of the repository.

Comment 7 The authors note: “Datavzrd stores data in JavaScript files, which are then loaded via static script tags.” This is an important implementation detail. However, the authors should also acknowledge that when data updates are needed, the visualiza- tions must be regenerated using the Datavzrd command line. This introduces some operational overhead, and its mention would provide a more balanced and transparent evaluation.

Response We have clarified in the manuscript that updating the input dataset requires regenerating the report:

This approach requires regenerating the report whenever the input dataset changes; how- ever, because data and configuration are completely separated, no updates to the latter are needed as long as the structure of the data remains unchanged. We recommend em- bedding the invocation of Datavzrd into automated workflows (e.g., using Snakemake, Nextflow , or CWL) to facilitate reproducible report generation after data updates.

Comment 8 The statement: “For the first time, this enables the collaborators and readers to interactively explore the data underlying the findings. . . ” is overly assertive. Without rigorous comparative analysis, claims of precedence should be avoided.

Response We agree and have removed the phrase “For the first time”.

Comment 9 Live Demo: https://datavzrd.github.io/example-report-moscot/ Suppl.%20Tab.%2013/index_1.html The Birth date field is interpreted as a string, preventing effective range-based searches. Additionally

---

## [Editor Report · Decision Letter 1]

8 Jul 2025

Datavzrd: Rapid programming- and maintenance-free interactive visualization and communication of tabular data

PONE-D-25-17640R1

Dear Dr. Wiegand,

Thank you for submitting the revised version of your manuscript, "Datavzrd: Rapid programming- and maintenance-free interactive visualization and communication of tabular data" to PLOS One. I have carefully reviewed the revisions alongside the reviewers' comments and am pleased to see that you have addressed the feedback thoroughly and thoughtfully. In light of the improvements made and the overall strength of the manuscript, I do not believe a second round of peer review is necessary.

We’re pleased to inform you that your manuscript has been judged scientifically suitable for publication and will be formally accepted for publication once it meets all outstanding technical requirements.

Kind regards,

Vivek Kumar, Ph. D.

Academic Editor

PLOS ONE
---

## [Editor Report · Acceptance letter]

PONE-D-25-17640R1

PLOS ONE

Dear Dr. Wiegand,

I'm pleased to inform you that your manuscript has been deemed suitable for publication in PLOS ONE. Congratulations! Your manuscript is now being handed over to our production team.

Kind regards,

on behalf of

Dr. Vivek Kumar

Academic Editor

PLOS ONE